# Repression of the Glucocorticoid Receptor Increases Hypoxic-Ischemic Brain Injury in the Male Neonatal Rat

**DOI:** 10.3390/ijms20143493

**Published:** 2019-07-16

**Authors:** Katherine R. Knox-Concepcion, Johnny D. Figueroa, Richard E. Hartman, Yong Li, Lubo Zhang

**Affiliations:** 1Lawrence D. Longo, MD Center for Perinatal Biology, Loma Linda University School of Medicine, Loma Linda, CA 92350, USA; 2Center for Health Disparities and Molecular Medicine, Loma Linda University School of Medicine, Loma Linda, CA 92350, USA; 3Department of Psychology, Loma Linda University, Loma Linda, CA 92350, USA

**Keywords:** glucocorticoid, glucocorticoid receptor, neuroprotection, Rice–Vannucci, neonatal, hypoxic-ischemic, encephalopathy, brain, neuroinflammation, functional outcomes

## Abstract

Hypoxic-ischemic encephalopathy (HIE) resulting from asphyxia is the most common cause of neonatal brain damage and results in significant neurological sequelae, including cerebral palsy. The current therapeutic interventions are extremely limited in improving neonatal outcomes. The present study tests the hypothesis that the suppression of endogenous glucocorticoid receptors (GRs) in the brain increases hypoxic-ischemic (HI) induced neonatal brain injury and worsens neurobehavioral outcomes through the promotion of increased inflammation. A mild HI treatment of P9 rat pups with ligation of the right common carotid artery followed by the treatment of 8% O_2_ for 60 min produced more significant brain injury with larger infarct size in female than male pups. Intracerebroventricular injection of GR siRNAs significantly reduced GR protein and mRNA abundance in the neonatal brain. Knockdown of endogenous brain GRs significantly increased brain infarct size after HI injury in male, but not female, rat pups. Moreover, GR repression resulted in a significant increase in inflammatory cytokines TNF-α and IL-10 at 6 h after HI injury in male pups. Male pups treated with GR siRNAs showed a significantly worsened reflex response and exhibited significant gait disturbances. The present study demonstrates that endogenous brain GRs play an important role in protecting the neonatal brain from HI induced injury in male pups, and suggests a potential role of glucocorticoids in sex differential treatment of HIE in the neonate.

## 1. Introduction

Hypoxic-ischemic encephalopathy (HIE) is a major cause of neonatal brain damage and occurs in three per thousand live term births and six per thousand premature infant births [1,2,3]. Each year in the United States, HIE affects 12,000 infants and causes 22% of neonatal deaths worldwide [4]. Currently, the standard of care for hypoxia-ischemia (HI) injury is hypothermia treatment, which has a limited capacity in reducing mortality and long-term negative neurodevelopmental outcomes [5]. Clinically, when neonates do not respond to this therapy, there are few candidates for alternative interventions [5]. Thus, there is an urgent need to develop new therapeutic techniques and adjuvant therapies to combat HI brain injury in infants.

The glucocorticoid receptor (GR) pathway has emerged as a promising target for the treatment of HI brain injury. GR plays a critical role in the development of infant birth weight, muscle tone, and its requirement for proper neuronal system maturation [6]. At birth, the GR is upregulated in the neonatal brain and peripheral organs, as it is necessary for organ maturation [7]. A large amount of evidence provides a close linkage between hypoxia and glucocorticoids. Neonatal hypoxia causes a surge in plasma adrenocorticotropic hormone (ACTH) and glucocorticoid levels, and long-term reprogramming of the HPA-axis [8,9]. We demonstrated that gestational hypoxia downregulated the GR in the developing brain via epigenetic mechanisms, rendering the neonatal brain vulnerable to hypoxic-ischemic injury by decreasing the GR-mediated neuroprotection [10,11]. Exogenous (i.e., therapeutic) and endogenous (i.e., physiologic) glucocorticoids play critical roles in the regulation of brain function and repair through activation of the GR [12]. Interestingly, studies of hypoxic-ischemic brain injury indicate that glucocorticoids provide both neuroprotective and neurotoxic effects [7]. The effect of glucocorticoids is influenced by the injection site, severity of injury, dosing, and timing of drug administration. Previous findings from our group showed that the administration of glucocorticoids conferred robust neuroprotection and ameliorated brain damage after severe HI injury in neonatal rat pups [13]. However, the role of endogenous GR in the immature brain in the pathological setting of HI injury remains unknown.

Clinically, HIE affects male infants more than female infants [14]. In the rodent model, brain damage is more severe in male rats affected by neonatal HI injury [15]. Interestingly, rodent models of HI injury do not show sex differentiation when the infarction size is severe [10,13,16,17,18]. Furthermore, hypothermic interventions targeted at HI injury protect female preferentially [15,16]. In hypoxic-ischemic injury, GR mRNA is preferentially increased in male neonatal rodents [10]. The role of glucocorticoids and sex-dependent protection in mild HI injury is currently unknown.

Herein, we investigate the role of GR in neonatal HI injury by knocking down endogenous brain GR in newborn rats. We examined the effects of GR repression on HI-induced inflammatory cytokine expression profile, acute brain injury, and short- and long-term neurobehavioral outcomes in an animal model of neonatal rats. We have shown that endogenous brain GRs play an important role in suppressing inflammatory cytokine production and protecting the neonatal brain from HI induced injury in male pups. The present findings suggest a potential role of glucocorticoids in sex differential treatment of HIE in the neonate.

## 2. Results

### 2.1. Knockdown of Glucocorticoid Receptor in the Neonatal Brain

To create an effective knockdown model, GR siRNAs (100 pmol) or the negative control were administered via intracerebroventricular (ICV) injection on postnatal day 7 (P7) of male and female rat pups. GR protein and mRNA abundance was measured 48 h after the treatment. As shown in Figure 1A, the GR siRNA treatment resulted in a significant decrease in GR protein abundance in the brain 48 h after the treatment. Consistently, RT-qPCR analyses showed that the GR siRNA treatment significantly downregulated GR mRNA abundance 48 h after the treatment (Figure 1B).

### 2.2. Effect of GR Repression on HI-Induced Infarction Size in the Neonatal Brain

In order to elucidate the role of GR in the neonatal brain during HI injury, we sought to create a mild HI injury model with ligation of the right common carotid artery followed by the treatment of 8% O_2_ for 60 min. As shown in Figure 2, the mild HI treatment produced greater brain injury with larger infarct size in female (13.49 ± 3.74 %, *n* = 8) than male (4.81 ± 2.80%, *n* = 5) pups. GR repression with siRNAs significantly increased HI-induced brain injury in male pups (12.96 ± 3.13%, *n* = 5 vs. 4.81 ± 2.80%, *n* = 5), but not in female pups (14.18 ± 2.56%, *n* = 11 vs. 13.49 ± 3.74%, *n* = 8) (Figure 2). Thus, the repression of endogenous brain GR minimized the difference in HI-induced infarction size in male and female pups (12.96 ± 3.13% vs. 14.18 ± 2.56%). Because GR repression showed no significant effect on HI-induced brain injury in female pups, we focused our following studies on male pups.

### 2.3. Effect of GR Knockdown on HI-Induced Changes in Short- and Long-Term Neurobehavioral Function in Male Pups

A battery of neurobehavioral tests were performed to examine the effect of GR repression on HI-induced short- and long-term function outcomes (Figure 3). We first evaluated the mortality of rat pups and weight gain after HI injury between animals that received scramble siRNAs as a negative control (negative control; 100 pmol) and GR siRNAs (100 pmol). There was no difference in the mortality rate between sham operated pups and HI-treated pups that received the negative control of siRNAs, however HI-treated pups with GR knockdown showed an increased mortality rate (Figure 4A). In the subsequent weight and behavioral data, we only include the data points from those animals that survived. We then measured the weight gain two, four, and six days after HI injury in male pups. Compared with the sham treatment, both groups of HI-treated pups that received the negative control or GR siRNAs decreased body weight gain (Figure 4B).

To assess post-HI motor and neurobehavioral deficits, we began with a battery of acute behavioral tests (negative geotaxis test, righting reflex test, and the wire hanging test) starting 48 h after HI injury and again at four days and six days after injury (Figure 3, Figure 4C–E). The negative geotaxis test assesses a reflex reaction initiated by vestibular and postural systems caused by the abnormal positioning of the head and body, requiring organized motor movement for successful completion [19]. Results from the geotaxis test showed that GR repression caused a significant time delay compared to the negative control group and sham animals in male pups (Figure 4C). Two-way ANOVA analysis indicated a main effect of group (F(2,95) = 6.538, *p* = 0.0022), days (F(2,95) = 10.94, *p* < 0.0001), and a group*days interaction effect (F(4,95) = 3.067, *p* < 0.05) of the latency in seconds for the geotaxis test. There were also significant differences two days after HI injury between treatment groups with GR siRNA versus the negative control group. Furthermore, the results showed that in the two-day window, the pups that experienced HI injury had significantly underperformed the sham group in response latency. The righting reflex test assesses simple motor coordination and is reflective of subcortical maturation. Two-way ANOVA analysis demonstrated a main effect of days (F(2,97) = 36.69, *p* < 0.0001) and group*days interaction effect (F(4,97) = 2.51, *p* < 0.0467) (Figure 4D). Two days after injury, repression of the GR during HI injury caused a significant increase in response latency compared to sham, indicating a worse performance. The wire hanging test evaluates both neuromuscular and locomotor functioning. Two-way ANOVA revealed a main effect of days (F(2,97) = 19.73, *p* < 0.0001) on latency to fall (Figure 4E). In the short-term studies, differences seen in performance between groups disappeared at four and six days after the initial HI insult.

Male rats were assessed again five, six, and seven weeks after HI injury with the CatWalk for sensorimotor deficits in order to quantify motor impairment (Figure 5). First, we found notable differences in the rats’ cadence, which describes the number of steps per second that the animal takes along its walking path (Figure 6A). Two-way ANOVA revealed a main group effect (F(2.99) = 9.324, *p* = 0.0002) and a time effect (F(2,99) = 5.948, *p* = 0.0036). Animals treated with GR siRNAs to repress the GR before HI injury performed significantly worse than the sham group at five, six, and seven weeks, whereas the animals with the negative control showed no significant difference from sham animals. As shown in Figure 6B, the run speed (cm/s) demonstrated a main group effect (F(2,99) = 4.561, *p* = 0.0127) and a main time effect (F(2.99) = 5.655, *p* = 0.0047). Animals that underwent HI-injury with GR siRNA treatment significantly performed worse than the sham group at seven weeks after the initial injury, whereas no significant difference was observed between animals with the negative control and sham group. We found a trend at the other time points, that GR repression caused worse motor impairments after HI injury than the non-treatment group.

The dynamic parameter, step cycle, examining the duration of the animals’ step and swing phase showed the motor changes between paws at five, six, and seven weeks after HI injury (Figure 6C). In general, there were no significant differences found between sham and animals received negative siRNAs with HI injury (Figure 6C). The right front paw demonstrated significant effects between time (F(2,66) = 8.152, *p* = 0.0007) and grouping (F(2,33) = 4.133, *p* = 0.0250) with a multiple comparisons test demonstrating significant differences in step cycle between the sham and GR siRNA with HI injury at seven weeks after injury. The left front paw had significant effects between the time (F(2,66) = 5.913, *p* = 0.0043) and grouping (F(2,33) = 5.537, *p* = 0.0085) with the multiple comparisons test showing significant differences at 5 weeks between sham animals and those treated with GR siRNA before HI injury. The right hind paw had significant time (F(2,66) = 8.485, *p* = 0.0005) and group (F(2,33) = 5.038, *p* = 0.0123) effects with significant differences between the sham and GR knockdown before HI at 5 and 7 weeks. Lastly, the left hind paw showed significant time (F(2,66) = 8.788, *p* = 0.0004) and grouping (F(2,33) = 5.322, *p* = 0099) effects with significant differences between the sham and GR siRNA treatment at five and six weeks after HI injury.

Stand is a static parameter of gait function and is described as the duration of contact of the paw with the glass. As shown in Figure 6D, the right front paw showed significant differences in both the time (F(2,66) = 9.297, *p* = 0.0003) and grouping (F(2,33) = 4.658, *p* = 0.0165) effects. The left front paw showed markedly significant differences in the time effect (F(2,66) = 10.61, *p* = 0.0001) and grouping effect (F(2,33) = 6.443, *p* = 0.0043). Two-way ANOVA analyses revealed significant changes in time of the left front paw contacting the glass during its gait between the sham and GR siRNA treatment groups at five, six, and seven weeks after HI injury. The right hind paw gave significant differences in time effect (F(2,66) = 14.02, *p* < 0.0001) and grouping effect (F(2,33) = 3.344, *p* = 0.047) with significant changes between the sham and GR siRNA treatment group 7 weeks after HI injury. Lastly, the left hind paw showed significant differences when considering the main time effect (F(2,66) = 14.1, *p* < 0.0001) and the grouping effect (F(2,33) = 4.269, *p* = 0.0224) with no significant changes in a follow-up analyses. Other parameters of max intensity, max area, print area, print width, print length, stride length, swing, duty cycle, and swing speed are described in detail in Appendix A.

At nine weeks, we performed the open-field test and Morris water maze (Figure 7). Interestingly, two-way ANOVA analyses revealed group (sham vs. negative control vs. GR siRNA) effects in the cued (F(2,33) = 4.211, *p* = 0.0235) and spatial learning day 1 (F(2,33) = 3.904, *p* = 0.031) without group effects evident in spatial learning day 2 and 3. By the third day of testing (spatial learning day 2), rats with HI injury were performing similar to their sham counterparts. In addition, we found noticeable differences in escape latency, the time an animal takes to reach the platform, without significant changes in the velocity of the animals. In the cued training, repression of the GR worsened the ability for the rats to escape, though they performed similar to their HI-injured counterparts by the two remaining trials. A two-way ANOVA showed significant group effects in the cued (F(2,33) = 7.193, *p* = 0.0026) and spatial learning day 1 (F(2,33) = 8.778, *p* = 0.0009). By spatial learning day 2 and spatial learning day 3, no significant differences were found. It is important to note that we found that the GR siRNA group appeared to perform worse during the probe trials. However, the behavioral performance during these trials was widely variable, which possibly contributed to the lack of significant differences between groups (Figure 7E,F). The total distance traveled in the open-field test did not show any significant group effects. Notably, HI injury alone caused a higher directional turn bias with repression of GR increasing this pattern in the open-field test (Figure 7L).

### 2.4. Effect of GR Knockdown on HI-Induced Changes in Inflammatory Cytokine Production in the Neonatal Brain

We further assessed the expression profile of key inflammatory cytokines in the neonatal rat brain *via* RT-qPCR 6, 12, and 24 h after HI injury. As shown in Figure 8, HI injury caused a time-dependent increase in mRNA abundance of pro-inflammatory cytokines TNF-α, IL-1b, IL-6, and anti-inflammatory cytokine IL-10. GR repression with siRNAs significantly increased HI-induced production of TNF-α and IL-10, and a tendency of increase of IL-1b at six hours. In contrast at 12 h after HI treatment, there was a tendency of decrease in TNF-α, IL-1b, and IL-10 in animals treated with GR siRNAs, as compared with those treated with the negative control. In addition, in sham groups at 24 h, there was a significant increase in TNF-α and a tendency of increase in IL-10 in animals treated with GR siRNAs, as compared with those treated with the negative control.

## 3. Discussion

This study demonstrates the following novel findings: One, repression of neonatal endogenous brain GR in a mild HI model sensitizes the neonatal brain to acute HI injury and results in increased brain infarction size in male neonatal rat pups; two, downregulation of brain GR causes greater impairment of short- and long-term neurobehavioral functioning after HI injury; and three, lastly, GR repression significantly increased HI-induced production of inflammatory cytokines TNF-α and IL-10 at six hours after HI injury. Thus, the present study provides evidence of a causal role of endogenous brain GR in the regulation of inflammatory response, acute brain injury and functional outcomes in the setting of hypoxic-ischemic injury. Of great interest, the effect of GR repression on mild HI injury in the neonate was a male specific change.

The goal of this study was to investigate the role of the GR in the pathogenesis of brain injury caused by asphyxia in near-term infants. We used P9 neonatal rats to closely mimic the brain development seen in a full-term newborn as a translational preclinical model [20,21,22]. A Rice–Vannucci model was implemented to evaluate the potential injury caused by a hypoxic-ischemic (HI) insult. In a full-term newborn, the GR is diffusely expressed throughout the whole brain and the expression level is highest at birth, when the HPA-axis becomes active and is critical to a newborn’s immediate survival [23]. The GR is vital for the normal central nervous system (CNS) development in the neonate and for the proper regulation of the HPA axis [12]. During hypoxic-ischemic encephalopathy (HIE), cerebral blood flow is compromised, causing widespread cellular dysfunction, neuroinflammation and apoptosis [7]. Previous studies have found that fetal hypoxia downregulates GR expression in the developing brain and decreases the GR-mediated neuroprotection in the neonatal brain in response to HI insult [10,11]. Furthermore, glucocorticoids are protective in a severe HI model of injury in both male and female neonatal rats [13].

In the present study, we sought to use siRNAs to knockdown GR expression in the neonatal brain to investigate the role of endogenous brain GR in HI-induced inflammation in mild brain injury and short- and long-term neurobehavioral outcomes. Our rationale was based on the complex regulation of glucocorticoid and progesterone receptor observed with other GR antagonist, including mifepristone [24,25]. We have shown that intracerebroventricular (ICV) injection of GR siRNAs significantly downregulated GR expression in the neonatal rat brain, providing a model to explore the role of endogenous brain GR in the pathogenesis of HI injury in the neonatal brain. Our previous studies showed that glucocorticoid administration via intracerebroventricular or intranasal injections is neuroprotective in both males and females in a severe HI model [13]. Evidence of both neuroprotective and neurotoxic effects exists in the literature and the effects seem to differ based on timing, dose, duration of treatment, and severity of injury [7,12]. The present finding that GR repression significantly increased mild HI-induced brain injury in male pups, without affecting female pups, is novel. Clinical findings mainly in preterm infants suggest a possible sex difference with a female advantage in long-term cognitive outcome [16]. However, currently there are no sex-specific follow-up data on HIE in term infants for meta-analysis due to the scarcity of research in this area. Because the pathophysiology and mechanism of HI brain damage in preterm versus term populations are quite different [16,26], it is of critical importance to investigate potential sex dimorphism in HIE of term brain injury as well. Some studies in rodent HIE models showed no significant difference between male and female pups in severe HI-induced brain injury of above 40% brain infarction in the ipsilateral hemisphere [10,13,16,17,18], although variable sex differences in long term cognitive outcome were observed [16]. A question of great interest is whether such a severe brain injury, which may not be often seen in the clinical setting, masks the subtle sex difference. The present study of a rat HIE model of term brain injury demonstrated a clear sex difference in HI-induced mild brain injury of around 5% brain infarction in male rat pups and 13% in female pup brains. Of importance, we found that transient knockdown of GRs in the neonatal brain by siRNAs eliminated the sex difference in acute brain injury caused by mild HI insult. These findings suggest an exciting and novel mechanism of endogenous brain GRs in the sex dimorphism of HIE in term infants.

We found that GR repression caused a higher mortality rate after HI injury. Previous studies showed that moderately dosed glucocorticoid treatment decreased the mortality rate and reduced the recovery time after HI injury [27,28,29]. Subsequently, our studies revealed that the knockdown of the GR decreased neurobehavioral performance in both short- and long-term neurobehavioral tests. The neurobehavioral outcomes of perinatal asphyxia are extremely important to evaluate because the sequelae of HI injury in children present as motor and cognitive deficits. The plasticity of the neonatal brain and the possibility of compensatory changes in the contralateral hemisphere make differences observed highly significant following HI injury. The Rice-Vannucci model is a widely accepted HIE model that has been used to study neurocognitive impairments that last into adulthood, such as cerebral palsy [30]. Given that chronic functional deficits and disability constitute a major sequela of neonatal HI injury, we served to explore the effect of GR repression on short- and long-term functional studies. Short-term neurobehavioral impairments are predictive of later functional impairments in patients that experienced neonatal asphyxia [31,32,33]. Moreover, because neurobehavioral studies have high variability, it is prudent to employ multiple behavioral tests to increase the specificity and sensitivity of the test protocol [34]. A recent study found that repression of the GR in the adult mouse brain caused worsened neurological performance after stroke [35]. Activation of the GR by glucocorticoids in the neonatal rat has had differing results in exacerbating or preventing long-lasting neurological impairments, though studies showing neurobehavioral outcomes are few [36,37,38].

We next observed the long-term neurobehavioral outcomes to improve our understanding of how the loss of endogenous GR may negatively impact brain recovery. We analyzed gait abnormalities with the CatWalk that revealed significant differences in the cadence, run speed, duration of the stance, and duration of the step cycle. Our study found that the pups with GR repression in the brain had a slower cadence or steps taken per second several weeks after HI injury. Interestingly, cadence is slower in stroke patients compared to their normal counterparts [39]. Similarly, cadence is impaired in rat models with intracerebral hemorrhage (ICH) [40]. Stand, or stance, is considered the duration of time the hind or front paws have ground contact in seconds. In the present study, the male pups with downregulated GR showed an increase in the amount of stand time, reflective of increased injury, principally in the opposing left front and right hind limbs. The bilateral impairment seen in these animals is due to compensatory postural adjustments for the injured limbs [41,42]. This is observed with the animal’s stance, with the hind limb opposite to the injured limb compensating for the front limb. The step cycle is described as the time in seconds between two consecutive contacts in the same paw and was significantly increased in animals with GR repression during the initial HI injury. A similar pattern presents in an adult middle cerebral artery occlusion stroke model [42].

The Morris water maze (MWM) is designed to assess learning and memory. In the present study, the animals that experienced HI injury demonstrated notable impairments in the distance travelled and time taken to reach the platform, without changes in confounding variables, such as velocity. Secondly, GR repression in the brain caused worsened performance outcomes than the other two groups, with improvement by spatial learning day 2. The cued learning is used as a control to safely determine whether the animals are performing similarly before testing for memory, a part of the test that is too often omitted in studies evaluating hypoxia-ischemia. The cued learning test requires motivation to escape from the water and consistency of rudimentary abilities of intact eyesight and locomotor ability that allows them to swim away from the wall and subsequently climb onto the platform [43]. Since the animals start at varying performance levels during the cued testing, it is difficult to interpret results on spatial learning days 1 to 3. Studies suggest that impairments in cued learning may be associated with changes in the striatum and not the hippocampus [44]. This observation is consistent with other studies that use the MWM as an accepted test to evaluate long-term changes from a hypoxic-ischemic insult, though the cued training data is not available [45,46]. Interestingly, we found a worsened performance of the animals that had downregulated GR at the time of injury in the latency to target, suggesting a deficit in either the locomotor ability to or the spatial memory required to reach the platform.

Neuroinflammation is a critical aspect of the pathogenesis of hypoxic-ischemic encephalopathy [7,26,47,48]. Neonatal encephalopathy is accompanied by an elevation of cytokines in the blood and cerebrospinal fluid [48,49]. Inflammation is a key aspect of the pathogenesis of HIE that may be manipulated to ameliorate further brain damage. The present study showed that mild HI insult produced time-dependent increases in pro-inflammatory cytokines TNF-α, IL-6, and IL-1β in the neonatal brain. Glucocorticoids are essential immunomodulatory agents, and microglial GRs act on several key processes, limiting pro-inflammatory actions of activated microglia in the brain [50,51,52,53,54]. We found that the knockdown of brain GRs with siRNAs caused an upregulation of pro-inflammatory cytokine TNF-α and anti-inflammatory cytokine IL-10 six hours after HI injury. In seeking a more clinically relevant model of mild HIE, these subtle changes detected may be the key to the exacerbated injury observed. In adult ischemia, TNF-α is decreased with glucocorticoid agonism and is connected to increased apoptosis [55,56]. Interestingly, the increase of IL-10 may be a compensatory response to the initial injury caused by GR repression. In addition, GR repression had a tendency to increase HI-induced IL-1β at six hours after HI treatment, but did not affect IL-6. These findings suggest GR’s complicated regulation of neural inflammatory response. It is possible that in the immature brain, downregulation of the GR allows specific targeting of a cellular subtype within the brain and thereby is specific to certain regions of the brain.

It is worth noting that the long-term results of dexamethasone and other glucocorticoids remain controversial, although in treatment of HIE, glucocorticoids acting on GR is a worthy candidate as an adjuvant therapy to hypothermia due to its broad ability to lower neuroinflammation and promote recovery after HI brain injury in the neonate. The present study revealed that GR plays a protective role in the neonatal brain. This is consistent with our previous findings that the administration of glucocorticoids before and after HI injury confers protection through the activation of the GR [10,13]. By knocking down the endogenous brain GR with siRNAs, the present study in a mild HI brain injury model demonstrated that GR repression increased infarction size, exacerbated negative neurobehavioral outcomes, and enhanced the inflammatory response. In Harding et al., we demonstrated no sex-specific differences in a severe HI-injury model [13]. Of great interest, mild HI-injury helped to reveal subtle sex differences, suggesting that male pup brains are better protected by the GR against mild HI-induced brain injury. Consistent with the present finding, a recent study demonstrated that females are more vulnerable with a preferentially lethal rate to genomic instability-induced inflammation during the early development, and males are protected by high levels of intrinsic testosterone [57]. The present study reveals GR action as another possible mechanism to protect males against inflammation induced by mild-HI injury in the male neonatal rat. Indeed, the previous study demonstrated that GR mRNA expression is significantly greater in male brains than that in female brains in both term fetuses and postnatal day 10 rat pups [10]. Similarly, GR mRNA abundance was found to be higher in the hippocampus and a tendency of greater levels in the cerebellum of male than female brains [58]. Thus, the present findings provide new insights into a potential therapeutic strategy of glucocorticoids as an adjuvant therapy in treatment of HIE with a sex preference in male infants. Follow-up studies are needed to simultaneously evaluate the female and male neonate to fully understand whether the behavioral changes and inflammatory markers detected are sex-dependent.

Several limitations of the study deserve to be mentioned. Our present study evaluated the negative impacts of GR repression in HI injury. To knockdown brain-specific GR, we used ICV injection, which is considered to directly impact the central nervous system. We cannot exclude the possibility of siRNA GR leakage to the peripheral system through the blood brain barrier and causing systemic effects and involvement of peripheral organ systems. The finding that GR repression exacerbated infarction size through TTC staining gave the macroscopic level of assessment of brain injury, though further approaches of brain MRI imaging and immunofluorescence may reveal more details in changes outside of infarction zones. The cytokine studies provided feasible biomarkers to understand the changes over time by which a mild form of HI injury. Future study of an immunohistochemical approach may provide additional information about the neuroinflammatory distribution of specific brain regions and the cell types that are involved in these differences.

This study provides evidence that GR may play a causal role in regulating acute brain injury, inflammation, and behavioral changes observed in mild HI injury. First, this study demonstrated that the repression of endogenous brain GR sensitizes the male neonatal brain to acute HI injury, as seen through an increase in brain infarction size. Secondly, we found greater impairment of short- and long-term neurobehavioral function after HI injury. Lastly, GR repression caused a significant increase of inflammatory cytokines TNF-α and IL-10 at six hours after mild HI injury in the male neonatal rat. This study provides evidence of the role GR may play in protecting the male neonatal brain from greater brain injury. Therapeutically, this study provides further evidence of the importance of cytokine profiles in HI-injury as biomarkers of injury severity.

## 4. Materials and Methods

### 4.1. Experimental Animals

Pregnant Sprague–Dawley rats were purchased from Charles River Laboratories (Portage, MI, USA). After the animals gave birth, pups were kept with dams in a room maintained at 24 °C with a 12-h light/dark cycle and provided ad libitum access to normal rat chow and filtered water. Pups of both sexes were randomly divided into groups with scramble control (negative control) or Nr3c1 (Glucocorticoid Receptor) siRNA (Dharmacon, Lafayette, CO, USA). All procedures and protocols were approved by the Institutional Animal Care and Use Committee (IACUC) of Loma Linda University (IACUC #:8160017, 6 April 2018) and followed the guidelines by the National Institutes of Health Guide for the Care and Use of Laboratory Animals.

### 4.2. Intracerebroventricular (ICV) Injection

Pups were anesthetized with 2% isoflurane in oxygen with loss of pedal reflexes as a measure of proper anesthetization. An incision was made on the skull to expose the skull in P7 neonatal rats using a stereotactic apparatus. We positioned a Hamilton syringe 2 mm inferior, 1.5 mm lateral, and 3 mm deep in relation to the bregma as referenced previously [13]. With the Hamilton syringe, 100 pmol of Nr3c1 siRNA (Dharmacon) or 100 pmol of Scramble siRNA was injected at 1 μl/min for 2 min. The syringe was held in the right cerebrum for an additional 3 min to prevent back-flow of siRNA. Once the Hamilton syringe was removed, the incision site was closed. The incision was sutured, anesthesia was removed, and the pups then recovered on a heating pad for 10 min before being returned to their dam.

### 4.3. Neonatal Hypoxic–Ischemic Encephalopathy (HIE) Rat Model

We performed HIE at P9 to reflect human brain development equivalent to a full-term infant [20,59,60]. A modified Rice–Vanucci rat model of hypoxic-ischemic encephalopathy was used as previously described [13]. In brief, P9 rat pups were anesthetized with 2% isoflurane in oxygen. Proper anesthetization was determined by loss of the pedal reflex. Pups were divided into sham and those undergoing HI injury. The right common carotid artery was ligated with a 5.0 silk surgical suture and cut between the ligations. Pups were then returned to dams. After one hour of recovery, pups were placed in a hypoxic chamber at 37 °C with 8% oxygen balanced with 92% nitrogen for 60 min. Pups were then returned to their dams to recover after hypoxic exposure. For the sham-treatment, pups were anesthetized with 2% isoflurane and the left carotid artery was exposed without ligation and without hypoxia treatment. Pups were subsequently returned to dams and allowed to recover.

### 4.4. Measurement of Infarction Size

Pups were euthanized 48 h after HI-injury to determine brain infarct size as previously described [10]. Serial coronal sections of the brain were cut (2-mm thick) and immersed it in a 2% solution of 2,3,5-triphenyltetrazolium chloride monohydrate (TTC; Sigma–Aldrich) for 5 min at 37 °C and fixed with 10% formaldehyde overnight. Both the caudal and rostral was photographed for subsequent analysis. The infarction size was analyzed by the Image J software (Version 1.40; National Institutes of Health, Bethesda, MD, USA), summed for each brain, and expressed as a percentage of the whole brain.

### 4.5. Neurobehavioral Tests

Neurobehavioral outcomes were evaluated after HI-injury with a battery of neurobehavioral tests. Assessments were performed at postnatal day 11, 13, and 15 (2, 4, and 6 days after injury) to evaluate short-term effects of the injury and interventions on subcortical maturation (righting reflex test), motor coordination and vestibular sensitivity (geotaxis test), and neuromuscular and locomotor development (wire hanging test). The chronic effect of the injury and interventions was evaluated by measuring locomotion (automated gait analyses), spatial learning and memory (Morris water maze), and general activity levels (open field test). *Reflex Test* was measured at P11, P13, and P15 pups. The pups were removed from dams and placed on a heating pad. The pups were placed on their back on a flat surface with forearms and hindlimbs being held in place. We recorded the amount of time taken for each pup to completely right itself with four paws on the surface. The average of three tests was recorded, with the overall group data expressed as response latency mean ± SEM. *Geotaxis Test* was measured at P11, P13, and P15 pups. The pups were placed with their heads pointing downward on a surface of approximately 30° incline. The surface had a cotton pad to provide traction for the pup. The time took to make a complete 180° turn was measured (time at completion = shoulders and head were facing upward to the slope). Each trial was performed for a maximum of 60 s. An average of three trials was calculated for each pup. The overall group data is expressed as response latency mean ± SEM. *Wire Hanging Test* was measured at P11, P13, and P15 pups. On P11, P13, and P15, pups were suspended from their forelimbs from a horizontal string between two metal rods (5mm × 5mm string, 47 cm long, and 50 cm high). After initiation of the time, pups were allowed to support themselves with their hind limbs to help prevent them from falling and allow them to traverse the string. An average of three trials was calculated for each animal. *CatWalk Test* was performed at 5, 6, and 7 weeks-post-HI injury. The CatWalk XT system is highly sensitive to locomotion and gait alterations following neurological injury (Noldus Information Technology Inc., Leesburg, VA, USA). Surprisingly, this automated test is rarely used in hypoxic-ischemic models to study gait abnormalities. In this test, the rats are allowed to walk across a glass walkway measuring 1.3 meters in length that is dimly illuminated with fluorescent light. Footprint images with intensity measurements are recorded by a camera positioned under the glass walkway. For our study, the rats were acclimated for 3 days before testing, with each animal given 10 min in a darkened goal box at the end of the runway and 5 min training on the runway. At the start of experiment, the rats were acclimated in the darkened goal box positioned at the end of the runway for 5 min. The animals were prompted to complete 5 compliant runs with 2 min between runs in the darkened box with their cage mate. Compliant runs are defined as runs lasting longer than 1 s but shorter than 15 s with a maximum allowed speed variation <60%. The CatWalk walkway was cleaned after three trials to ensure accurate data collection. Footprint identification and labeling was performed using the CatWalk XT software (CatWalk XT v10.6, Noldus Information Technology Inc., Leesburg, VA, USA) and static and dynamic parameters generated for 5 compliant runs per rat. The mean score from 5 trials was analyzed. *Morris Water Maze* was performed at 9-weeks post injury. The rats were evaluated for spatial learning and memory function using the standard Morris water maze (MWM) test. The MWM paradigm consisted of a three-day procedure including cued learning (day 1) and spatial learning (day 2 and 3). Probe trials were taken at the beginning of day 2 and 3 for spatial memory evaluation [61]. The rats were placed in a metal pool (100 cm diameter) filled with water. The animals were allowed to swim to an escape platform that was either 1.5 cm above the water surface during the cued trial, or 1.5 cm below the water surface during the spatial learning or absent from the water during the probe trials. Each animal was administered 10 trials per day (60 s max per trial) in 5 blocks of 2 consecutive trials. *Open-Field Test* was performed at 9-weeks post injury. The open-field test (OFT) was performed in the rats at 9 weeks after the HI injury. The maze was constructed from metal with a dark base surface (maze dimensions, 49 cm long, 35.5 cm wide, 44.5 cm tall). Light levels in the room were dimmed with halogen lights at a distance to provide ample light. The rats were observed in the OFT for 30 min and movement parameters were recorded by an overhead camera and analyzed by a computerized tracking system (Noldus Ethovision; Information Technology, Inc., Leesburg, VA, USA).

### 4.6. Real time RT-qPCR

Total RNA was extracted from the right hemisphere whole brain tissue 6, 12, and 24 h after hypoxic-ischemic insult. Total RNA was extracted using the TRIzol reagent (Thermo Fisher Scientific, Waltham, MA, USA) with subsequent reverse transcription using the iScript cDNA synthesis system (Bio-Rad, Hercules, CA, USA). RNA quality was assessed using a Nanodrop spectrophotometer (Thermo Fisher Scientific) by determining A_260_/A_280_ and A_260_/A_230_ values. RNA used had a value of A_260_/A_280_ value of 1.9–2.0 and A_260_/A_230_ value of 2.0–2.3. We then performed subsequent electrophoresis of 1 μg of total RNA on a 1% agarose gel to reveal intact 28S, 18S, and 5S RNA species. We then measured mRNA abundance of GR, TNF-α, IL-10, IL-6, and IL-1β using iQ SYBR Green Supermix (Bio-Rad). The following reverse transcription polymerase chain reaction protocol was used: 95 °C for 5 min followed by 40 cycles of 95 °C for 15 s, 60 °C for 1 min. The GR primers used were 5’-aggtctgaagagccaagagtt-3’ (forward) and 5’-tggaagcagtaggtaaggaga-3’ (reverse). The TNF-α primers used were 5’-gccgatttgccacttcatac-3’ (forward) and 5’-aagtagacctgcccggactc-3’ (reverse). The IL-10 primers used were 5’-cactgctatgttgcctgctcttac-3’ (forward) and 5’-gggtctggctgactgggaag-3’ (reverse). The IL-6 primers used were 5’-gcctattgaaaatctgctctgg-3’ (forward) and 5’-ggaagttggggtaggaagga-3’ (reverse). The IL-1β primers used were 5’-agcaacgacaaaatccctgt-3’ (forward) and 5’-gaagacaaaccgcttttcca-3’ (reverse). PCR was performed in triplicate, and threshold cycle numbers (C_T_) was generated by CFX connect Real Time System (Bio-Rad) were averaged for each sample. Internal reference was glyceraldehyde-3-phosphate dehydrogenase.

### 4.7. Statistical Analysis

The data were expressed ± standard error of the mean (SEM). Experimental number (*n*) represents pups from multiple dams. The data was assessed by a two-way analysis of variance (ANOVA) followed by a Holm–Sidak post-hoc test for comparisons of multiple groups or Student’s t-test (unpaired, two-tailed) for comparisons between two groups, where appropriate using the Graph-Pad Prism software (GraphPad Software Version 7, San Diego, California, CA, USA). For all comparisons, *p* < 0.05 indicated statistical significance.

## Figures and Tables

**Figure 1 ijms-20-03493-f001:**
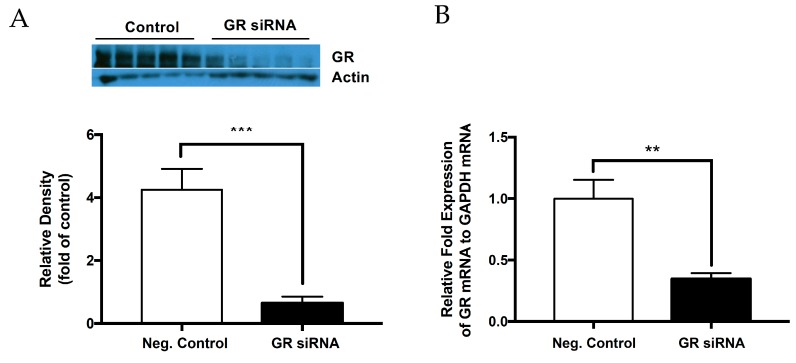
Glucocorticoid receptor is effectively knocked down at the protein and mRNA level. The glucocorticoid receptor (GR) siRNA repressed GR expression in the neonatal rat brain. The neonatal rats received either GR siRNA (100 pmol) or negative control (100 pmol) via intracerebroventricular (ICV) injection seven days after birth (P7). The brain samples were collected 48 h after ICV treatment and glucocorticoid receptor (GR) protein abundance was evaluated via Western blotting (**A**) and mRNA abundance was evaluated (**B**) via real-time RT-qPCR relative to GADPH, respectively. Data are means ± SEM. *n* = 5. ** *p* < 0.01 *** *p* < 0.001, GR siRNA vs. negative control.

**Figure 2 ijms-20-03493-f002:**
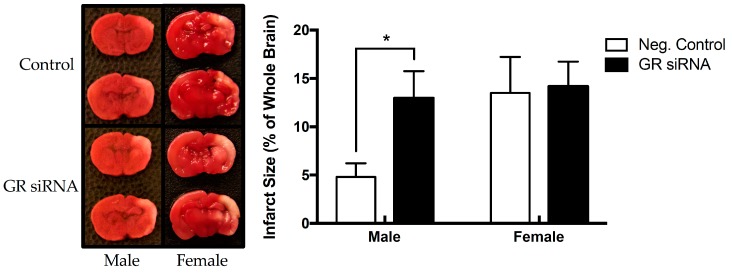
Knockdown of endogenous GR exacerbates mild HI injury in the male neonatal rat pup brain. GR siRNA (100 pmol) or negative control (100 pmol) was injected by ICV injection 48 h on postnatal day 7 (P7) before HI-injury. HI-injury was performed on postnatal day 9 (P9) for 60 min in 8% FiO_2_. We separated sex into the following groups: negative control (negative control: male, *n* = 5; female, *n* = 8) versus GR siRNA (male, *n* = 5; female, *n* = 11). In the male population, GR siRNA increased infarct volume. No significant difference was seen in the female population, although the total brain injury was increased with a mild hypoxic-ischemic model. Data are means ± SEM, * *p* = 0.0315, GR siRNA vs. negative control.

**Figure 3 ijms-20-03493-f003:**
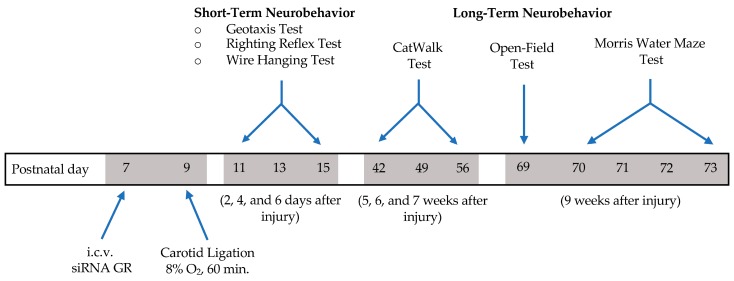
Design/timeline of short- and long-term behavioral tests after hypoxic-ischemic injury. The neonatal male rats received GR siRNA (100 pmol) or negative control (100 pmol) by ICV injection on postnatal day 7 (P7). Animals were then divided into two groups, Sham and HIE. Hypoxic-ischemic injury was performed on postnatal day 9 with carotid ligation and subsequent 8% O_2_ exposure for 60 min. Animals in sham group had carotid arteries exposed without subsequent hypoxic-ischemic injury induction. HI-injury was performed on postnatal day 9 (P9) for 60 min in 8% FiO_2_. Pups were returned to dams. Neurobehavioral tests began two days after injury and extended to 9 weeks after initial injury.

**Figure 4 ijms-20-03493-f004:**
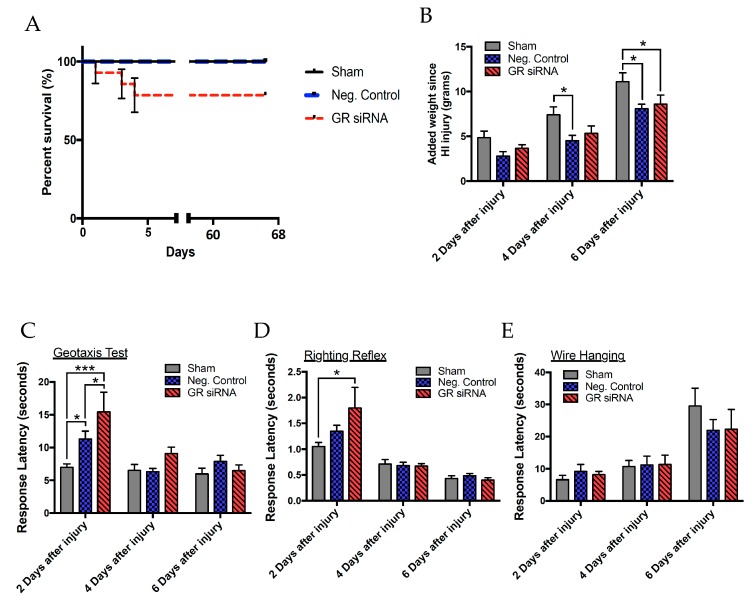
Repression of GR worsens short-term neurobehavioral deficits in the neonatal male rat with mild HI injury. The neonatal male rats received GR siRNA (100 pmol) or negative control (100 pmol) by ICV injection 48 h before HI-injury (8% O_2_; 60 min). The sham group did not undergo HI-injury treatment. The short-term neurobehavioral assessment was taken two, four, and six days after HI insult. (**A**) Mortality rate with Groups: negative control without injury (sham); (*n* = 12) versus negative control with HI injury (*n* = 13) versus GR siRNA (*n* = 14) with HI injury. Subsequent weight and behavioral studies do not include animals that did not survive. (**B**) Weight added since HI injury, (**C**) geotaxis test, (**D**) righting reflex, and the (**E**) wire hanging test were evaluated accordingly. Groups for weight and short-term behavior were negative control without injury (sham); (*n* = 12) versus negative control with HI injury (*n* = 13) versus GR siRNA (*n* = 11) with HI injury. Data are means ± SEM, * *p* < 0.05, *** *p* < 0.0001.

**Figure 5 ijms-20-03493-f005:**
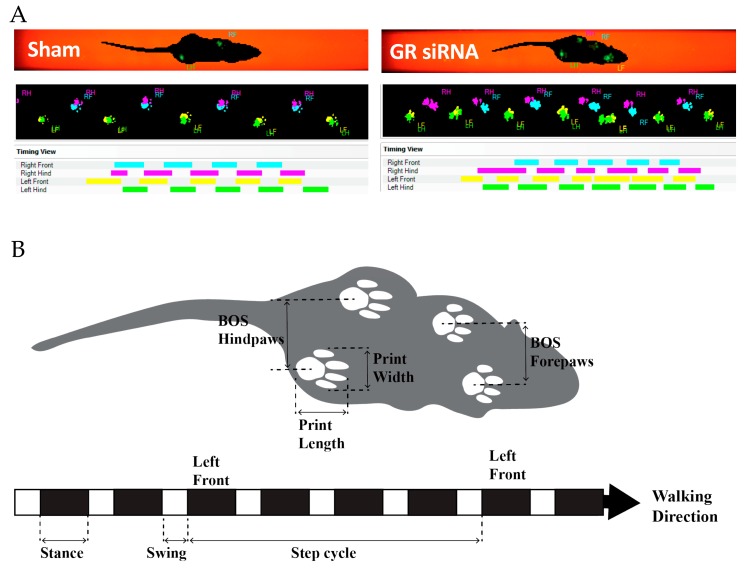
Schematic illustration of the CatWalk gait parameters. The CatWalk is a highly sensitive tool that analyzes the gait and locomotion of rodents. A camera detects the illuminated footprints and weight distribution on a glass plate. The animal traverses the glass plate to a goal box. (**A**) Two walking patterns were collected at five weeks post HI injury with GR siRNA treatment or sham. (**B**) A pictorial representation of the CatWalk parameters. The black boxes represent the stance, which is the duration of time a paw is in contact with the glass. The white boxes represent the swing duration of a paw when it is not in contact with the glass. A step cycle describes the time, in seconds, from when an initial paw contact on the glass to the next time the paw comes in contact with glass. Additional data is included in Appendix A.

**Figure 6 ijms-20-03493-f006:**
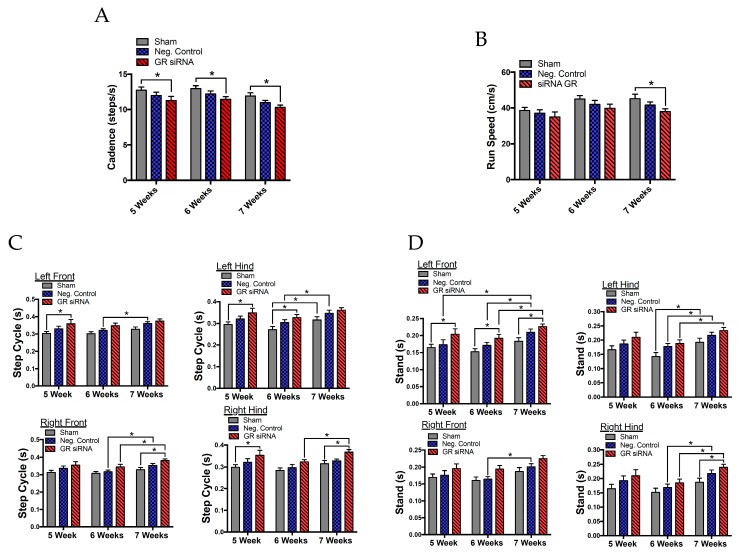
Repression of GR worsens long-term neurobehavioral deficits in the neonatal male rat with mild HI injury. The neonatal male rats received GR siRNA (100 pmol) or negative control (100 pmol) by ICV injection 48 h before HI-injury (8% FiO_2_; 60 min). Groups were sham (negative control; (*n* = 12) versus negative control with HI injury (*n* = 13) versus GR siRNA (*n* = 11) with HI injury. Catwalk test was performed at five, six, and seven weeks after HI injury for (**A**) cadence, (**B**) run speed, (**C**) step cycle duration, and (**D**) stand duration. Data are means ± SEM, * *p* < 0.05.

**Figure 7 ijms-20-03493-f007:**
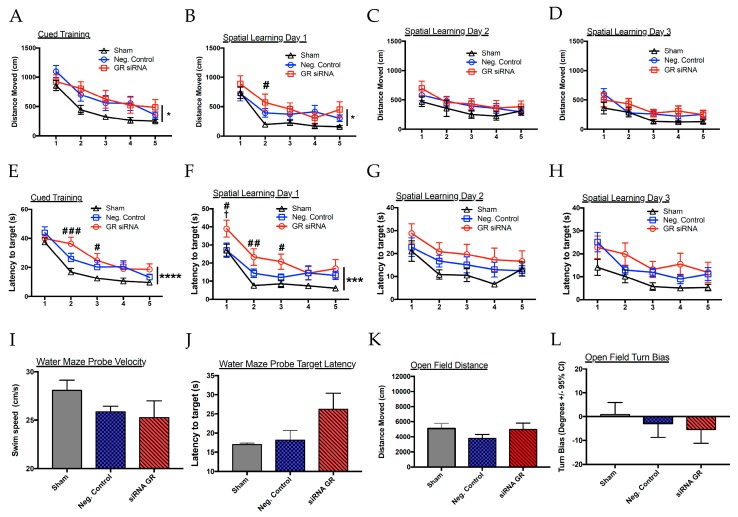
Repression of GR and long-term neurobehavioral deficits in the neonatal male rat with mild HI injury. The neonatal male rats received GR siRNA (100 pmol) or negative control (100 pmol) by ICV injection 48 h before HI-injury (8% FiO_2_; 60 min). Groups were sham (negative Control; (*n* = 12) versus negative control with HI injury (*n* = 13) versus GR siRNA (*n* = 11) with HI injury. (**A**–**D**) Morris water maze cued training to spatial learning day 3 testing for the distance moved before arrival to the platform. (**E**–**H**) Cued training to spatial learning day 3 for the latency to platform in seconds. (**I**,**J**) Probe trials performed before Spatial 2 and 3. (**K**,**L**) Open-field test. * *p* < 0.05, *** *p* < 0.01, **** *p* < 0.001, two-way ANOVA group (sham vs. negative control vs. GR siRNA) effect. # *p* < 0.05, ## *p* < 0.01, ### *p* < 0.001 sham *vs.* GR siRNA. † *p* < 0.05, negative control *vs.* GR siRNA.

**Figure 8 ijms-20-03493-f008:**
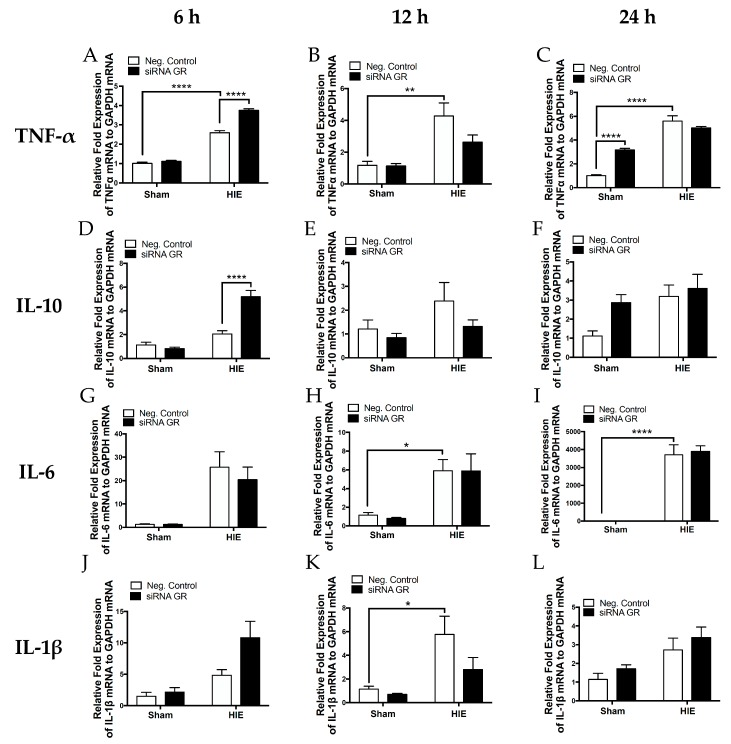
Repression of the GR increases cytokine profile 6 h after mild HI injury. The neonatal male rats received GR siRNA (100 pmol) or negative control (100 pmol) by ICV injection 48 h before HI-injury (8% FiO_2_; 60 min). Groups were sham with negative control, sham with GR siRNA, negative control with HI injury, and GR siRNA with HI injury. Whole brain RNA was isolated at six, 12, and 24 h after HI injury and cytokines TNF-α (**A**–**C**), IL-10 (**D**–**F**), IL-6 (**G**–**I**), and IL-1β (**J**–**L**). * *p* < 0.05, ** *p* < 0.01, **** *p* < 0.0001.

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
