# Peer review of "Repression of the Glucocorticoid Receptor Increases Hypoxic-Ischemic Brain Injury in the Male Neonatal Rat"

_ijms, 2019, doi:10.3390/ijms20143493_

Round 1

Reviewer 1 Report

International Journal of Molecular Sciences (ijms-536112)

Repression of the Glucocorticoid Receptor Increases Hypoxic-Ischemic Brain Injury in a Sex-dependent Manner in Male Neonatal Rats

      This manuscript presents a study of the effects of glucocorticoid receptor repression on hypoxic-ischemic encephalopathy (HIE) in neonatal rat pups, with attention to sex differences.

      The authors used a model of mild HIE in P9 rat pups. With this model female rats demonstrated a 2-3-fold greater infarct size than male pups. The authors then used either a siRNA directed against the glucocorticoid receptor (GR), a scrambled siRNA, or a sham. Decrease in GR expression in male rats worsened the effects of the HIE model, increasing the infarct size to that of females, and worsening survival. The authors then used a variety of behavioral tests (geotaxis test, righting reflex, wire hanging, catwalk test, open field testing, Morris water maze) and cytokine concentrations in the brain (TNF-a, IL-1b, IL-6, IL-10). Performance on the geotaxis test and righting reflex was worsened at 2 days after HIE in male rats receiving GR siRNA. A variety of gait parameters were worsened at timepoints between 5 and 7 weeks in male rats receiving GR siRNA after HIE. Spatial learning in the Morris water maze was impaired at learning day 1. GR siRNA also increased the inflammatory cytokines TNF-a and IL-10 at 6 hours after HIE, but not at 12 or 24 hours.

      The authors conclude that there findings add data to the complicated story of the role of steroids in HIE, with interactions between sex and severity of injury which are not fully understood.

      Overall I feel the manuscript is of interest, but it would benefit from attention to several points.

1.    The brain sections in Fig 2 seem more rostral for the males than the females.

2.    A lot of data is presented in this study and the field is complicated. The first paragraph of the Discussion could be improved for clarity (e.g. line 272 “enhances the impairment” is awkward) and a final paragraph attempting to clearly state the impact these findings have would help to present the data more strongly.

Author Response

Reviewer 1

1.      The brain sections in Figure 2 seem more rostral for the males than the females.

Response: The images shown in Figure 2 are representative slides of each subgroup. The final infarction size result is based on a percentage of the whole brain from analyzing six serial coronal sections. 

2.      A lot of data is presented in this study and the field is complicated. The first paragraph of the Discussion could be improved for clarity (e.g. line 272 “enhances the impairment” is awkward) and a final paragraph attempting to clearly state the impact these findings have would help to present the data more strongly.

Response: Line 279 is changed to “causes greater impairment”.  As suggested, we added a final paragraph (Line 428 to 436): “This study provides evidence that GR may play a causal role regulating acute brain injury, inflammation, and behavioral changes observed in mild HI-injury. This study demonstrated that repression of brain endogenous GR sensitizes the male neonatal brain to acute HI injury, as seen through an increase in brain infarction size. Secondly, we found greater impairment of short- and long-term neurobehavioral function after HI injury. Lastly, GR repression caused a significant increase of inflammatory cytokines TNF-α and IL-10 at six hours after mild HI injury in the male neonatal rat. This study provides evidence of the role GR may play in protecting the male neonatal brain from greater brain injury. Therapeutically, this study provides further evidence of the importance of cytokine profiles in HI-injury as biomarkers of the severity of injury.”

Reviewer 2 Report

In the manuscript by Knox-Concepcion et al., authors reports the involvement of glucocorticoid receptor on behavioral , neuroinflammatory, histological and molecular levels outcomes using pre-clinical neonatal HIE rat model. The study is well-designed and clearly describes obtained results in both in vivo and in vitro setups. The manuscript is well written and contains important information that might have potential clinical impact.

Minor comments:

The expression of GC is not limited to the brain and can be found in other tissues including muscle, blood, kidney, etc. The i.c.v. injected siRNA might be leaky to circulation, especially keeping in mind that neonatal rat pups were used,  and by that affect not only brain but also, for example muscle expression of GR. This, in turn, may affect neuro-motor behavior outcomes. Did authors considered this possibility or, based on previous experience, it was excluded? Please discuss at least.

Line 228: potential misprint “…takes to teach the platform…” – maybe “.. to reach the platform”

Lines 236 and 239: couldn’t find Figures 5e-f and Figure 5h. Please correct.

In Figures 4, 6, 8 the stars, showing the significance, are too small. Likewise in Fig 6, the OX axis’ text is too small. Please enlarge.

Author Response

Reviewer 2

1.      The expression of GC is not limited to the brain and can be found in other tissues including muscle, blood, kidney, etc. The i.c.v. injected siRNA might be leaky to circulation, especially keeping in mind that neonatal rat pups were used, and by that affect not only brain but also, for example muscle expression of GR. This, in turn, may affect neuro-motor behavior outcomes. Did authors considered this possibility or, based on previous experience, it was excluded? Please discuss at least.

Response: Line 426: Added “To knock-down brain-specific GR, we used i.c.v. injection, which is considered to directly impact the central nervous system. We cannot exclude the possibility of siRNA GR leakage to the peripheral system through the blood brain barrier and causing systemic effects and involvement of peripheral organ systems.”

2.      Line 228: potential misprint “…takes to teach the platform…” – maybe “.. to reach the platform”

Response: This is corrected. Thank you for catching this. On Line 230, we changed “…takes to teach the platform…” to “…takes to reach the platform…”

3.      Lines 236 and 239: couldn’t find Figures 5e-f and Figure 5h. Please correct.

Response: On line 238: Figure 5e-5f changed to Figure 7e-7f.  On line 244: Figure 5h changed to 7l.

4.      In Figures 4, 6, 8 the stars, showing the significance, are too small. Likewise in Fig 6, the OX axis’ text is too small. Please enlarge.

Response: Figure 4, 6, and 8: increased the stars showing significance and axis text to ensure the figures are readable. Thank you for pointing this out.